# *"You don't want anyone who hasn't been through anything telling you what to do, because how do they know?"*: Qualitative analysis of case managers in a hospital-based violence intervention program

Hannah C. Decker[1]*, Gwendolyn Hubner[1], Adaobi Nwabuo[1], Leslie Johnson[2], Michael Texada[1], Ruben Marquez[1], Julia Orellana[1], Terrell Henderson[1], Rochelle Dicker[3], Rebecca E. Plevin[1], Catherine Juillard[3]

1 Department of Surgery, University of California at San Francisco, San Francisco, California, United States of America, 2 Rollins School of Public Health, Emory University School of Medicine, Atlanta, Georgia, United States of America, 3 Department of Surgery, University of California at Los Angeles, Los Angeles, California, United States of America

* hannah.decker@ucsf.edu

**Data Availability Statement:** Data cannot be shared publicly because of the sensitive nature of

## Abstract

### Statement of purpose

Intentional violent injury is a leading cause of disability and death among young adults in the United States. Hospital-based violence intervention programs (HVIPs), which strive to prevent re-injury through intensive case management, have emerged as a successful and cost-effective strategy to address this issue. Despite the importance of strong therapeutic relationships between clients and their case managers, specific case manager behaviors and attributes that drive the formation of these relationships have not been elucidated.

### Methods

A qualitative analysis with a modified grounded theory approach was conducted to gain insight into what clients perceive to be crucial to the formation of a strong client-case manager relationship. Twenty-four semi-structured interviews were conducted with prior clients of our hospital's HVIP. The interviews were analyzed using constant comparison method for recurrent themes.

### Results

Several key themes emerged from the interviews. Clients emphasized that their case managers must: 1) understand and relate to their sociocultural contexts, 2) navigate the initial in-hospital meeting to successfully create connection, 3) exhibit true compassion and care, 4) serve as role models, 5) act as portals of opportunity, and 6) engender mutual respect and pride.

the interviews that delve into issues of immigration, violence, and criminal persecution. However, codebooks, code structures, and de-identified manuscripts can be made available upon request to Sue Peterson (sue.peterson@sfdph.org) for researchers who meet the UCSF Institutional Review Board's criteria for access to confidential data.

**Funding:** The author(s) received no specific funding for this work.

**Competing interests:** The authors have declared that no competing interests exist.

## Conclusions

This study identifies key behaviors of case managers that facilitate the formation of strong therapeutic relationships at the different stages of client recovery. This study's findings emphasize the importance of case managers being culturally aligned with and embedded in their clients' communities. This work can provide a roadmap for case managers to form optimally effective relationships with clients.

## Introduction

Intentional injury is a leading cause of disability and death among young adults in the United States, particularly in minority populations [1, 2]. Non-fatal assaults outstrip fatal assaults by roughly 89:1 and account for significant healthcare utilization and societal costs [3–6]. Up to 45% of violently injured youth sustain a second violent injury, making prior violent injury one of the strongest predictors of future violent injur [5, 7–10].

Hospital-based violence intervention programs (HVIPs) operate on the premise that violent injury is a potent incentive for behavioral change and work to prevent reinjury by addressing root causes of violence [11–17]. The Wraparound Project (WAP) is a HVIP based at Zuckerberg San Francisco General (ZSFG), San Francisco's only Level 1 Trauma Center. WAP offers services that seek to address the challenges violently injured patients face following hospital discharge, including finding safe housing, accessing mental health services, returning to work or school, obtaining medical care, battling legal issues, and others [14]. WAP is a model for fledgling HVIPs across the country, as it has been shown to be both effective and cost-effective [14, 18].

Strong therapeutic relationships between clients and their case managers are key to the success of HVIPs. WAP, like many HVIPs, employs a peer-based model for hiring case managers [17]. Despite their importance, the specific case manager behaviors, attributes, and actions that help form these relationships have not been elucidated; this study explores client perceptions of case managers to better understand them. As HVIPs become a more widespread violence prevention tool, these results can suggest a roadmap for case managers, particularly at fledgling HVIPs, to develop effective client relationships.

## Methods

### Qualitative approach and research paradigm

We employed qualitative analysis to capture the complexity of participant experiences, with a modified grounded theory approach to gain insight into the important relationships between WAP's clients and their case managers [19–22]. We followed the Consolidated Criteria for Reporting Qualitative Research guidelines for reporting the qualitative analysis performed [23].

### Context and sampling strategy

The primary researchers were two female, Caucasian fourth-year medical students (HD and GH) trained in trauma-informed interviewing. Interviews took place at WAP offices or over the phone.

Eligible participants completed WAP's six-month period of intensive case management and were over 18 years old. We used a purposive sampling strategy, utilizing the case managers

as gatekeepers to former WAP clients, a difficult to reach population [24] Participants were contacted in-person or by telephone. We recruited both typical cases (e.g. clients who maintained a neutral to positive relationship with their case managers) and intense variants (e.g. clients with uniquely strong relationships with their case managers), to illustrate a range of examples of the relationship of interest [24] HD and GH attended shared research progress at WAP staff meetings, which helped identify missing perspectives (e.g. female or reinjured participants).

We used the concept of information power to assess the sufficiency of our sample size, taking into account the study's relatively narrow breadth, sample specificity, strategy for analysis, and, most critically, the quality of the dialogue [25] Quality of the dialogue was particularly important given the interviewers were culturally discordant with the interviewed persons, though received training on trauma-informed interviewing and embedded themselves in the daily workings of the HVIP. A multidisciplinary research group, including three trauma surgeons, two medical students, one injury prevention professional, and four case managers determined when sufficient information power was achieved by evaluating the richness of the data. The final two interviews were completed after information power was deemed sufficient to ensure no novel themes were uncovered.

### Data collection methods

GH conducted the first 7 of the interviews; HD conducted the others. The interviews averaged 40 minutes and ranged from 10 to 68 minutes. One interview was conducted in Spanish with the aid of an in-person interpreter. Interviews were guided by a series of questions (S1 Appendix) developed by GH and HD and informed by the literature surrounding HVIPs and the experience of WAP case managers. Data collection and analysis occurred concurrently so that any new themes that emerged could be probed.

In-person interviews were recorded with Otter, a voice-note iPhone application that securely stores data in the Cloud. Phone interviews were recorded with Call Recorder, an iPhone application that securely encrypts and stores telephone conversations on Amazon S3 servers. Hand-written field notes (S2 Appendix) were kept to track demographic information, contextual details, and key points in the interviews.

### Sample characteristics

Twenty-four (24) former Wraparound clients were interviewed. Of these, 14 identified as African American or Black, 8 as Latino/a, and 2 as other races or ethnicities. Twenty-one (21) identified as men and 3 identified as women. Of the 24 participants, 23 were fluent in English and 1 spoke only Spanish. Participant ages averaged 26 years and ranged from 19–35 years. Participants were interviewed after the official six-month period of intensive case management had concluded, though several were still engaged in case management services to varying degrees. A comparison to WAP's overall demographics is shown in Table 1.

### Data processing & analysis

Interviews were transcribed and anonymized by the interviewer and transcripts were stored on a password-protected computer. Audio from two interviews was lost to recording failures. A memo was written immediately following the interview to capture as much of the information as possible.

A codebook was created from six transcripts chosen for their depth and diversity, using an inductive approach to generate codes. Codes were assigned to discrete incidents, ideas, or dynamics. The resulting codebook was discussed with the research team to ensure conceptual

**Table 1. Comparison of participant and WAP overall demographics.**

|  | *Study Sample* | *WAP Overall* |
|---|---|---|
|  | *N(%)* | *N(%)* |
| Gender |  |  |
| Male | 21 (88) | 394 (86) |
| Female | 3 (13) | 65 (14) |
| Race/Ethnicity |  |  |
| Black/African American | 14 (58) | 215 (47) |
| Latino | 8 (33) | 200 (44) |
| White | 0 (0) | 23 (5) |
| Other | 2 (8) | 21 (5) |
| English Proficiency |  |  |
| Yes | 23 (96) | 229 (50) |
| No | 1 (4) | 229 (50) |
| Median Age | 26.5 | 21 |

clarity. Three transcripts were dually coded by the two interviewers and discrepancies were discussed and resolved. The rest of the transcripts were then coded as they were transcribed. Units of text were assigned to codes using Atlas.ti [26].

De-identified interview transcripts were analyzed using constant comparison method as outlined by Glaser and Straus [21]. Blocks of text were compared to similar content, both within and between transcripts and groups of transcripts. Clusters of codes were condensed into themes, and transcripts were re-visited through the lens of these themes. These themes were discussed among the research team which allowed for new conceptual development and further thematic refinement.

### Ethical issues pertaining to human subjects

The Institutional Review Board at UCSF specifically approved this study. All participants gave informed consent to participate in the recorded interviews and received $20 gift cards. All participants were given the autonomy to take breaks or stop the interview if the conversation became emotionally overwhelming. A mental health specialist was available on site if immediate support was needed.

## Results

Participant interviews unearthed key themes about case manager attributes and behaviors. While the relationship between case managers and their clients is often non-linear, the interviews highlighted that certain themes were more important at different stages of the relationship (Fig 1). These were: 1) understand and relate to client sociocultural context, 2) navigate in-hospital teachable moments to create connection with clients, 3) exhibit true compassion and care, 4) act as role models, 5) serve as portals of opportunity, and 6) engender mutual respect and pride.

### Theme 1: Understand and relate to client sociocultural context

Nearly all participants found it important that their case manager understand their sociocultural context, including community ties, economic conditions, and violence. Participants exhibited varied preferences for the way case managers gained this understanding (Fig 2). Some appreciated that their case manager had similar life experiences. As one client said, "how

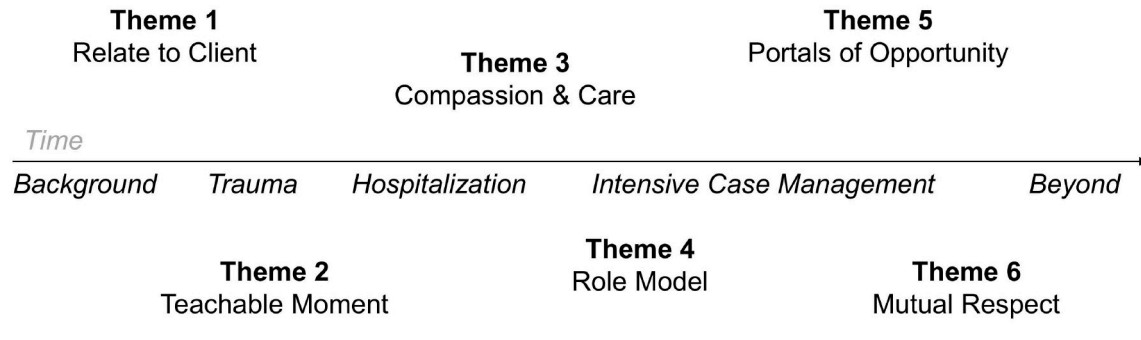

**Fig 1. Key themes by WAP case management time course.**

is a person who. . .only reads a book—how can they relate to . . . this kid that just got stabbed 50 times in the street?" Some clients voiced they were more open to advice from case managers who had similar life experiences. As one client said, "You don't want anyone who hasn't been through anything telling you what to do, because how do they know?"

Many clients, more often African-American men, expressed a desire to share a racial and gender identity with their case manager. One client said, "I couldn't tell you that I would have opened up to somebody that wasn't a minority. . . at the time. Because everybody look like the police!" Almost all participants who identified as Latino/a appreciated that their case manager spoke Spanish fluently, as this facilitated better communication with the client and their families. The importance of language extended beyond the use of formal language and included the ability to "code switch" with clients. One participant described this phenomenon: "You can't really connect with someone unless you can really talk to them at their level."

In terms of gender, some male participants found their case manager to be an important role model of masculinity. No female participants voiced a preference for a female case manager (though all women interviewed had a male case manager).

Some participants did prefer that their case managers had similar demographics or life experiences; however, most participants cared more that their case managers were open to learning about participants' lives without judgment. This perspective was more common among female participants and participants who did not identify as African-American. One noted, "You don't have to go through that stuff to know what it's like. Just put yourself in their shoes."

## Theme 2: Navigate in-hospital teachable moments

The initial meeting in the hospital following injury is a critical moment in the case manager-client relationship for case managers to build trust. Participants near-universally described hospitalization as a stressful experience. Case managers can build early rapport by

| Open to learning, listening without judgment | Exposure through past work or integration with community | Shared demographics | Shared life experiences |

*No preference for shared attributes* *Strong preference for shared attributes*

**Fig 2. Participant preferences for mechanism by which case managers develop sociocultural understanding.**

validating how frightening trauma and hospitalization can be. Additionally, many participants were initially suspicious of unknown people coming into their room; they appreciated when case managers identified themselves immediately and explicitly denied affiliation with law enforcement.

Case managers also developed trust with clients by engaging with their friends and family members. This was particularly important if clients were sedated during their initial hospitalization and case managers connected with family members during that time. One participant said, "I woke up out of the coma and . . . my mama was like you need to talk to [your case manager] because he's gonna help you. And. . . I trusted my mom." For other participants, familiarity with case managers from the community or through mutual social connections was key to building trust at the outset.

Another action prioritized by clients during hospitalization was when case managers delivered on key promises early. Further, case managers can continue to strengthen relationships during hospitalization by visiting clients, even briefly, after "everyone else had to go back to their lives."

## Theme 3: Exhibit true compassion and care

Participants emphasized the importance of believing case managers truly cared about them, and highlighted key behaviors that reinforced this belief (Fig 3).

Spending time getting to know clients helped demonstrate that case managers sincerely desired to help, and weren't "doing it for a paycheck." The perception that case managers would not "give up on them" was similarly important. One client stated:

*"I started smoking meth . . . And instead of [my case manager], you know, talking bad to me, or telling me to get out. . . he sat there sat outside with me. . . Not once did I see judgment in his eyes. . .When most people would have got mad on me and looked down on me, you know, he brought me in closer and let me know that everything was gonna be alright."*

Participants appreciated their case manager going "the extra mile" to provide services by driving them to medical appointments, sitting with them in court, or finding them housing through their personal network. These actions made participants feel as though case managers wanted to help them achieve their goals, rather than just "checking a box." For many participants the longevity of the participant-case manager relationship was another indication of their case manager's commitment. One client said, "That's love, you know. . .He been there through the tough times, the good times, and all that. He just been poppin' up on me, just showing up."

Showing up in the hospital: asking clients how they are, appreciating the impact trauma can have on life

Getting to know clients on a personal level, going the extra mile while providing services, ensuring outcomes, not giving up on clients during setbacks

Being dependable, accessible and reliable over the long term

*Trauma*   *Hospitalization*   *Intensive Case Management*   *Beyond*

**Fig 3. Case manager behaviors clients perceive as showing they care, over time.**

## Theme 4: Act as role models

One dynamic many participants highlighted was the position of WAP case managers in the landscape of mentors and role models. Potential formal and informal role models that were discussed included "OG" community members, other case managers, physicians, and law enforcement (Fig 4).

For some participants, older community members who have successfully avoided or overcome violence, legal issues, and substance use were key role models, but other participants pointed to potential pitfalls of listening to community members. One indicated: "that man that's out there on the block putting a gun in your hands is not wanting to see you do good." Participants pointed to hypocrisy as one reason not to trust some community members, with one explaining: "You know, like I got OGs who—they say, 'Man, you shouldn't do drugs.' And then five minutes later, they're around the corner smoking weed with everybody on the block. Or they tell you, 'Man . . . you shouldn't be out here gangbanging.' And then they go around the back of the car tire and put their pistol on the back of they pants."

Formal mentors can also pose challenges. Several participants described case managers from other programs who had not completely severed ties with an old way of being, with one person describing a former case manager who asked to buy drugs from him. A number of participants felt that the doctors treating them following their trauma stereotyped them and were uncaring. One participant said, "The doctors like—they wouldn't even ask me, you know, like how I'm doing. It's all 'How'd you get shot? Are you a gang member?'"

Case managers were described as sitting at the intersection of formal and informal mentors, in many cases serving as both personal and professional role models. As one client put it, "From the way he. . .deals with his own children. . .to the way he deals with kids out in the street, to the way he deals with his own clients. . . that's the kind of man I want to look up to."

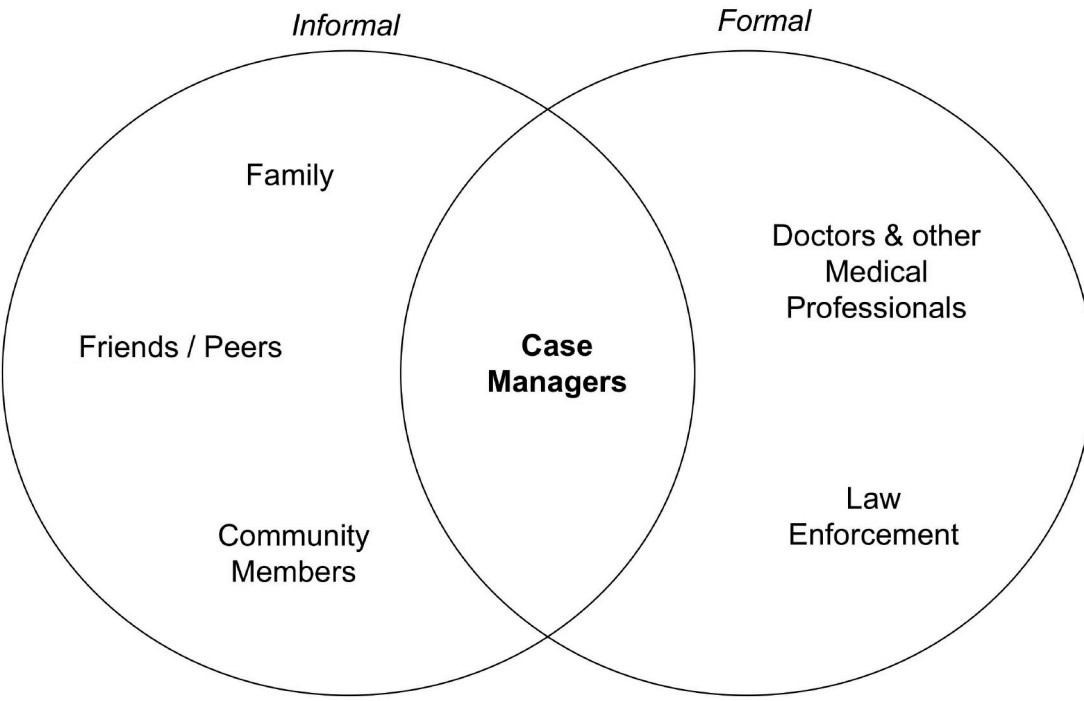

**Fig 4. Case managers sit at the intersection of formal and informal potential mentors.**

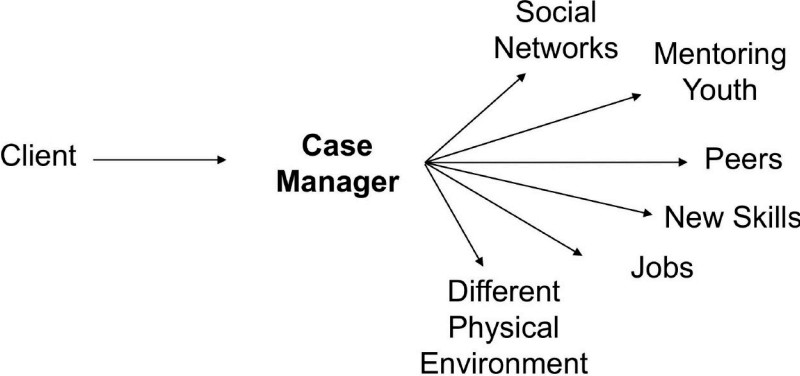

**Fig 5. Case managers as portals of opportunity for clients.**

### Theme 5: Serve as portals of opportunity

Case managers serve as portals of opportunity to connect participants to new social networks, economic opportunities, and physical spaces (Fig 5). For some, these new networks replaced those that reinforced behaviors keeping them entrenched in the cycle of violence. These new opportunities also provided an alternate source of income and pride and a way to stay busy, which was critical to avoiding re-injury. As one client put it, he "just needed something to keep me from having the extra time on my hands 'cause the neighborhood that I live in is basically what caused me to be a victim."

### Theme 6: Engender mutual respect and pride

By demonstrating and requiring respect, WAP case managers engendered a sense of pride and responsibility in participants. One appreciated that his case manager "talks to me like an adult, like a friend" because "other people talk to me like I'm a little kid." Interestingly, one younger participant expressed the opposite, noting that case manager-type figures in his past had treated him as too much of an adult, expecting, for example, that he knew more than he did about the intricacies of the legal system. He appreciated his WAP case manager's ability to joke, smile, and break down complex concepts for him. The importance of balancing treating clients as equals while providing support was highlighted in multiple interviews.

Participants noted that case managers expected effort from clients commensurate with their abilities. One participant described falling out of contact with his case manager. His case manager said, "I'm not about to place nothing in your hands. You'll have to work for it." The client appreciated this, saying "I don't want no handouts. I want somebody that is gonna make me go get it." Participants frequently cited "tough love" as a way case managers showed that they respected their clients, with multiple participants remarking that case managers "are not gonna sugar coat it."

This mutual respect helps empower clients to choose to change their lives. Placing the responsibility on clients to engage with WAP highlights an important reality described by multiple participants: to benefit from WAP services, "you have to be ready." As one participant put it, "I never became the person that I. . . was trying to be until I actually wanted to fully be that person."

## Discussion

This study underscores the centrality of strong client-case manager relationships to clients' success in WAP. Clients perceive certain case manager characteristics, behaviors, and

identities as essential to the strength of these relationships. Participants stressed that their case managers must understand their social and cultural context, as this enhanced trust and credibility. Clients highlighted that there are myriad ways case managers can develop this important understanding: from being enmeshed in the community where violence is commonly occurring, to experiencing violence first-hand, to sharing a racial and ethnic background with clients, to simply listening without judgment. This supports the practice among HVIPs of hiring case managers who are culturally concordant with the client population, an approach that aligns with a push to provide culturally competent care throughout medicine [27] Of note, this practice can often present a challenge to HVIPs embedded in institutions that may have strong biases against individuals who have interacted with the justice system, including restrictions against hiring individuals with prior felony convictions. Cultural competence is a key strategy to reducing disparities in health care and modestly improves health outcomes and patient satisfaction in culturally and linguistically diverse patients [28, 29] WAP clients also confirmed the importance of case manager behaviors that can be learned and taught such as exhibiting compassion, acting as role models, providing opportunities, and engendering mutual respect.

To our knowledge, this is the only qualitative analysis of HVIPs to date that gives clients the opportunity to share their experiences with case managers in their own words. These thoughts are particularly important as HVIPs are expanding across the country, as they can enhance program exportability by providing guidance to HVIPs when hiring and coaching case managers [27] This study suggests that HVIPs should prioritize cultural concordance and community embeddedness when hiring case managers. This requires a deep understanding of not only the demographics of the at-risk population, but also the communities plagued by violence. HVIPs should also prioritize case manager behaviors that are key to developing strong therapeutic relationships.

The contrast many clients highlighted between case managers and other healthcare providers underscores that all individuals interacting with victims of violence in the healthcare system should be educated in trauma-informed care. Case managers at WAP are trained to provide trauma-informed care, but also know it implicitly, and participants appreciate these behaviors.

One limitation to this study is that qualitative analysis generates themes, ideas, and questions rather than definitive conclusions. In addition, this study captured experiences from participants in a single HVIP in a single city, so the generalizability of the themes explored may be limited. Given the interactive nature of qualitative interviews, as well as the importance of tone and body language, these interviews were primarily conducted in English, with only one non-English speaking client included. Though over half of WAP's clients over the past decade have been English speaking, there may be needs of non-English speaking clients that were not captured in this study. The culture and gender of the interviewers was not concordant with most of the clients, which also may have affected the depth of the interviews especially around sensitive topics such as interaction with law enforcement, mental health, immigration, and more.

Almost no participants were willing to expand on what, if any, factors had negatively impacted their relationships with their case managers—even when pressed. This may reflect the significant emotional potency of these relationships. This also may be due to the gatekeeper strategy employed to recruit participants—the case managers may have preferentially recruited clients with whom they had positive relationships, and clients may only have been willing to interview if they had a positive experience to share. Additionally, even though all clients had completed the intensive period of case management following the initial trauma, some were still engaged in case management services to varying degrees. This involvement may have influenced participant responses, even though it was explicitly stated that their participation or

lack thereof in interviews would have no bearing on services provided and all conversations would be anonymized prior to being shared.

## Conclusion

This study provides an opportunity to hear in clients' own words what drives strong relationships with HVIP case managers and gives case managers at fledgling programs a roadmap for developing optimally effective client relationships.

## Supporting information

**S1 Appendix. Interview guide.**
(DOCX)

**S2 Appendix. Post interview contact summary sheet.**
(DOCX)

## Author Contributions

**Conceptualization:** Hannah C. Decker, Gwendolyn Hubner, Adaobi Nwabuo, Leslie Johnson, Michael Texada, Ruben Marquez, Terrell Henderson, Rochelle Dicker, Rebecca E. Plevin, Catherine Juillard.

**Data curation:** Hannah C. Decker, Gwendolyn Hubner.

**Formal analysis:** Hannah C. Decker, Gwendolyn Hubner, Adaobi Nwabuo.

**Investigation:** Hannah C. Decker, Gwendolyn Hubner, Adaobi Nwabuo.

**Methodology:** Hannah C. Decker, Gwendolyn Hubner, Adaobi Nwabuo, Leslie Johnson, Rochelle Dicker, Rebecca E. Plevin, Catherine Juillard.

**Project administration:** Hannah C. Decker, Michael Texada, Ruben Marquez, Julia Orellana, Terrell Henderson.

**Supervision:** Rochelle Dicker, Rebecca E. Plevin, Catherine Juillard.

**Visualization:** Hannah C. Decker.

**Writing – original draft:** Hannah C. Decker.

**Writing – review & editing:** Gwendolyn Hubner, Adaobi Nwabuo, Leslie Johnson, Michael Texada, Ruben Marquez, Julia Orellana, Terrell Henderson, Rochelle Dicker, Rebecca E. Plevin, Catherine Juillard.

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
