## [Decision Letter · Decision Letter 0]

17 Feb 2020

PONE-D-19-21048

“You don’t want anyone who hasn’t been through anything telling you what to do, because how do they know?”: Qualitative Analysis of Case Managers in a Hospital-Based Violence Intervention Program

PLOS ONE

Dear Dr. Decker,

Thank you for submitting your manuscript to PLOS ONE. After careful consideration, we feel that it has merit but does not fully meet PLOS ONE’s publication criteria as it currently stands. Therefore, we invite you to submit a revised version of the manuscript that addresses the points raised during the review process.

We would appreciate receiving your revised manuscript by Apr 02 2020 11:59PM. To enhance the reproducibility of your results, we recommend that if applicable you deposit your laboratory protocols in protocols.io, where a protocol can be assigned its own identifier (DOI) such that it can be cited independently in the future. For instructions see: http://journals.plos.org/plosone/s/submission-guidelines#loc-laboratory-protocols

We look forward to receiving your revised manuscript.

Kind regards,

Janhavi Ajit Vaingankar

Academic Editor

PLOS ONE

Journal Requirements:

Reviewers' comments:

Reviewer's Responses to Questions

**Comments to the Author**

1. Is the manuscript technically sound, and do the data support the conclusions?

Reviewer #1: Yes

Reviewer #2: Yes

Reviewer #3: Yes

2. Has the statistical analysis been performed appropriately and rigorously? 

Reviewer #1: N/A

Reviewer #2: I Don't Know

Reviewer #3: Yes

3. Have the authors made all data underlying the findings in their manuscript fully available?

Reviewer #1: No

Reviewer #2: Yes

Reviewer #3: Yes

4. Is the manuscript presented in an intelligible fashion and written in standard English?

Reviewer #1: Yes

Reviewer #2: Yes

Reviewer #3: Yes

5. Review Comments to the Author

Reviewer #1: The authors have conducted a thorough qualitative research study assessing the key points of strong, effective, therapeutic relationships between case managers and clients in the Hospital-based Violence Intervention Program setting. Although they used only one site to recruit patients, they included a sizeable number of interviewees which could feasibly reach theoretical saturation, or information power. While this is the first study of its kind, their results are not altogether surprising, and as presented, do not necessarily add much to the existing knowledge on this topic. I would recommend that the authors resubmit the manuscript after including more illustrative quotations from their interviews, and also attempting to generate conclusions which might contribute to the theoretical knowledge base on their topic. This might be done by extrapolating theoretical insights from their findings, or else, by relating their findings more specifically to existing literature. There is a rich amount of literature around therapeutic alliance, and it deserves reference within the context of this type of study. The authors note that their raw data is not available to the public; this is an incontrovertibly reasonable position to take given the social precarity of many of their research subjects. Overall it is an excellent study, but requires just a bit more work to make it an interesting and useful contribution to the field.

Line 78 – The authors should comment more specifically on what elements of information power were considered, and how the attributes of the interviewers were considered in this assessment.

Line 133 – At what point in time were the clients in the sample seen by WAP? Were they actively engaged in case management, or had the relationship ceased, and for how long? Can you comment on how this temporal aspect may or may not have changed your results?

Line 257 – I am curious to know more about what factors can negatively impact the therapeutic relationship; Perhaps you can summarize this within a new/different theme, or at least address it more fully within the context of the themes you’ve already reported.

Line 260 – The length of the discussion is a bit cursory; the authors should attempt to make a contribution to theory, or at least to tie their results to more literature surrounding therapeutic relationships, case management, and trauma.

Line 274 – I would also add that, attempting to hire case managers with similar backgrounds can often present a challenge to HVIPs embedded in institutions that have strong biases against individuals who have interacted with the justice system, often times with restrictions against hiring individuals with prior felony convictions. Yet the need for cultural similarity is so paramount that hospitals should reconsider these restrictions.

Line 282 – Please expand on the potential impact of the language issue on your conclusions, and also on what the implications of the attributes of the interviewers are on your results.

Figure 4&5 – I don’t find that these figures add much to your study. My preference would be to instead include a table with more illustrative comments from each of the themes.

Reviewer #2: I was pleased to review the manuscript, “You don’t want anyone who hasn’t been through anything telling you what to do, because how do they know?”: Qualitative Analysis of Case Managers in a Hospital-Based Violence Intervention Program” for publication in PLOS ONE. This research study was a qualitative analysis of the characteristics of case managers in hospital-based violence intervention programs (HVIPs) that maximize success when working with patients.

This study is particularly timely since HVIPs have expanded significantly over the past decade, including a push from the American College of Surgeons Committee on Trauma that tacitly nudges trauma centers to begin such programs. Despite this, there remains a large gap in knowledge, which this study helps to fill.

Overall, I found the content and results of the study to be a solid addition to the research base. Regarding the specifics on research methodology, I must defer to the advice of other reviewers as I have limited expertise in qualitative research/grounded theory methodology.

In general, I have minor revisions to suggest, which I will detail below:

1. Page 4, line 70: You report that participants “completed WAP’s period of intensive case management…” Can you simply quantify how long this period is?

2. Page 5, line 95: “Hand-written field notes were kept to track demographic information, contextual details, tone and key points in the interview.” Since the appendix provided doesn’t have any questions specific to tone, I would reword this. I’m sure tone may have been captured by the reviewers, but since it wasn’t specifically included, it’s uncertain if this was always reported.

3. Page 7, line 136: I have been debating in my head the validity of Figure 1 as presented. In the current presentation, it portrays the relationship between patient and case manager as progressing with time in a linear fashion from theme 1 to theme 6. Although this is a useful schema to think about the concept, my experience is that most patients progress in a non-linear fashion throughout time and the importance of different themes may come and go during the course of treatment. Unless the underlying data clearly shows that different themes were more important at different time periods, I would either remove this Figure or redesign it.

4. Pages 7-11: One point that struck me on multiple occasions was the use of the phrase “some participants” to describe themes. Since the phrase was used on many occasions, it left me wondering if all of those themes were brought up at roughly the same amount or if the themes were actually brought up with varying frequency (a small number of participants vs half vs most). Given the small sample size of this study, I do think that the use of the phrase is understandable, but if it is possible/appropriate to utilize words that describe the relative magnitude, it would be helpful for the reader. For example, if I were hiring for a case management position, I’d be very interested to know the intensity with which clients valued similar life experiences. Depending on that answer, it might change my hiring priorities.

5. Pages 9-10: The segment outlining Theme 4 could use additional details. As it reads now, it predominately discusses the risks of informal mentors and the challenges sometimes faced with formal mentors. The segment would benefit from additional details on why participants highlighted the importance of role models apart from the pitfalls of other information sources. A selected quote might be helpful.

6. General thoughts on supplemental materials: Overall, I found the figures/graphs/appendices to be very helpful. With the exception of Figure 1 (discussed above), I thought the inclusion of the others were helpful in making concrete otherwise abstract concepts.

Thank you for your work on this important topic.

Reviewer #3: This is a well-designed, well-executed qualitative analysis of opinions of participants in an HVIP on preferable characteristics of HVIP case managers. These were: 1) understand and relate to client sociocultural context, 2) navigate in-hospital teachable moments to create connection with clients, 3) exhibit true compassion and care, 4) act as role models, 5) serve as portals of opportunity, and 6) engender mutual respect and pride.

Overall, this is a much needed patient-centered analysis on the qualities of an HVIP case manager that best serve clients. Many hospitals provide services to victims of violence (e.g. victim’s assistance offerings, police reporting at the bedside, information about support groups, medical coverage, and follow-up appointments for injuries). These services may be delivered by myriad staff (e.g. registration, technicians, clerical staff, nurses, social workers, physicians). However, delivering true trauma-informed care to victims of inner city violence requires someone to establish trust, have a shared understanding of life, show real empathy, and be dedicated and altruistic enough to continue their support after the hospital visit ends. This analysis not only provides a window into understanding the features of an HVIP case manager that are preferred by victims of violence, but also helps inform on how should be delivering services to these patients in the hospital setting.

Capitalizing on these teachable moments and forming a meaningful connection during the profound moment of injury can help prevent injury recidivism. However, too often we use ‘conventional’ outcomes such as re-injury or recidivism within the criminal justice system to measure success of HVIPs, failing to seek the perspective of those we serve. Although their cost-effectiveness has been shown, HVIPs alone cannot eliminate the social determinants of health that lead to violent injuries. However, HVIPs are an important part of the puzzle. What our clients feel matters and their connections expressed here with their case managers is positive and appreciative, providing hope for an all too often disenfranchised population. It is common sense that fearful patients would want “someone to talk to them on their level,” and that they would want someone who could put themselves in their shoes. Giving participants new opportunities for education, jobs, training, treatment, and safe physical spaces naturally builds participant resiliency and pride. With racial tensions often high in American cities, it is not surprising that case managers who share similar demographic features were often preferred by clients, but more important (particularly among female participants), was that case managers were open-minded and really cared. These features speak to how trust is gained. Participants want to be understood, just like every person. In fact, this study informs all of us on how to care for victims of violence, and I agree with the authors that “all individuals interacting with victims of violence in the healthcare system should be educated in trauma-informed care.”

Please note recommended edit below.

168 Some participants did not prefer that their case manager share THEIR demographics or life experiences, but wanted

169 THEM to be open to learning about THEIR life context without judgment. (Misplaced modifiers: unclear who these pronouns refer to--the participants or their case managers.)

Recommend changing to: "Some participants did prefer that their case managers had similar demographics or life experiences. Rather, participants cared more that their case managers were open to learning about participants' lives without judgment."

6. PLOS authors have the option to publish the peer review history of their article (what does this mean?). If published, this will include your full peer review and any attached files.

Reviewer #1: No

Reviewer #2: Yes: Kyle R Fischer

Reviewer #3: Yes: Dr. Katherine M Bakes

---

## [Author Response · Author response to Decision Letter 0]

25 Apr 2020

Please see cover letter with response to reviewers.

---

## [Decision Letter · Decision Letter 1]

1 Jun 2020

“You don’t want anyone who hasn’t been through anything telling you what to do, because how do they know?”: Qualitative Analysis of Case Managers in a Hospital-Based Violence Intervention Program

PONE-D-19-21048R1

Dear Dr. Decker,

We are pleased to inform you that your manuscript has been judged scientifically suitable for publication and will be formally accepted for publication once it complies with all outstanding technical requirements.

With kind regards,

Janhavi Ajit Vaingankar

Academic Editor

PLOS ONE

Additional Editor Comments (optional):

Reviewers' comments:

Reviewer's Responses to Questions

**Comments to the Author**

1. If the authors have adequately addressed your comments raised in a previous round of review and you feel that this manuscript is now acceptable for publication, you may indicate that here to bypass the “Comments to the Author” section, enter your conflict of interest statement in the “Confidential to Editor” section, and submit your "Accept" recommendation.

Reviewer #1: All comments have been addressed

Reviewer #2: All comments have been addressed

2. Is the manuscript technically sound, and do the data support the conclusions?

Reviewer #1: Yes

Reviewer #2: (No Response)

3. Has the statistical analysis been performed appropriately and rigorously? 

Reviewer #1: Yes

Reviewer #2: (No Response)

4. Have the authors made all data underlying the findings in their manuscript fully available?

Reviewer #1: No

Reviewer #2: (No Response)

5. Is the manuscript presented in an intelligible fashion and written in standard English?

Reviewer #1: Yes

Reviewer #2: (No Response)

6. Review Comments to the Author

Reviewer #1: The comments have been responded to adequately. I appreciate the commentary on gatekeeper strategy used by the case managers.

Reviewer #2: (No Response)

7. PLOS authors have the option to publish the peer review history of their article (what does this mean?). If published, this will include your full peer review and any attached files.

Reviewer #1: Yes: Erik Kramer

Reviewer #2: Yes: Kyle Fischer

---

## [Editor Report · Acceptance letter]

5 Jun 2020

PONE-D-19-21048R1 

*“You don’t want anyone who hasn’t been through anything telling you what to do, because how do they know?”:* Qualitative Analysis of Case Managers in a Hospital-Based Violence Intervention Program 

Dear Dr. Decker:

I'm pleased to inform you that your manuscript has been deemed suitable for publication in PLOS ONE. Congratulations! Your manuscript is now with our production department. 

Kind regards, 

on behalf of

Ms Janhavi Ajit Vaingankar 

Academic Editor

PLOS ONE